# Control of electronic topology in a strongly correlated electron system

Sami Dzsaber[1], Diego A. Zocco [1], Alix McCollam[2], Franziska Weickert [3], Ross McDonald [3], Mathieu Taupin[1], Gaku Eguchi[1], Xinlin Yan[1], Andrey Prokofiev [1], Lucas M. K. Tang[2], Bryan Vlaar[2], Laurel E. Winter [3], Marcelo Jaime [3], Qimiao Si [4] & Silke Paschen [1] ✉

It is becoming increasingly clear that breakthrough in quantum applications necessitates materials innovation. In high demand are conductors with robust topological states that can be manipulated at will. This is what we demonstrate in the present work. We discover that the pronounced topological response of a strongly correlated "Weyl-Kondo" semimetal can be genuinely manipulated—and ultimately fully suppressed—by magnetic fields. We understand this behavior as a Zeeman-driven motion of Weyl nodes in momentum space, up to the point where the nodes meet and annihilate in a topological quantum phase transition. The topologically trivial but correlated background remains unaffected across this transition, as is shown by our investigations up to much larger fields. Our work lays the ground for systematic explorations of electronic topology, and boosts the prospect for topological quantum devices.

Key to uncovering the richness of phenomena displayed by strongly correlated electron systems and to understanding the underlying mechanisms has been the great tunability of these systems, which enabled a systematic exploration of the landscape of their low-temperature phases[1–4]. As an example, such studies led to the discovery that high-temperature superconductivity is in many, if not all cases an emergent phase stabilized by quantum critical fluctuations[5,6], and that critical electron delocalization transitions[7,8] may play an important role therein[9]. Such deep insight is indispensable to tailoring properties at will, and ultimately exploiting them for applications.

Electrons are strongly correlated if their mutual Coulomb repulsion reaches or exceeds the same order as their kinetic energy. In bulk materials, this regime is typically realized by means of $f$ and/or $d$ orbitals with a suitable strength of orbital overlap. Flat bands, renormalized by orders of magnitude compared to simple metals and pinned to the Fermi energy, can be achieved via the Kondo effect, driven by a spin exchange interaction between localized (typically $4f$) and itinerant ($s$, $p$, $d$) electrons[10]. Recently, it has been demonstrated that such very strongly correlated conductors may also exhibit extreme signatures of nontrivial topology[11]. This raises the question as

to whether the excellent tunability in terms of correlation physics can also be exploited to control topology per se.

Our work shows that this can indeed be achieved. Upon tuning $Ce_3Bi_4Pd_3$[12], a Weyl-Kondo semimetal[11–13], by magnetic field we observe a continuous suppression of the giant topological response associated with the material's Kondo-driven Weyl nodes, and the annihilation of the nodes. This transition is characterized by the suppression of singularities in the Berry curvature instead of a Landau order parameter; as such, it is a topological quantum phase transition. An important and surprising aspect is that this transition happens in an only smoothly varying correlated background; one could instead have expected topological states to be essentially inert to tuning parameter changes. This discovery opens up a new regime in the correlation–topology interplay, and will facilitate a systematic search for new correlation-driven topological phases.

## Results and discussion

Before presenting our results and explaining them in detail, we summarize the key signatures of the Weyl-Kondo semimetal $Ce_3Bi_4Pd_3$ in zero magnetic field[11,12] (see Supplementary Note 1 for details). They are

[1]Institute of Solid State Physics, Vienna University of Technology, 1040 Vienna, Austria. [2]High Field Magnet Laboratory (HFML-EMFL), Radboud University, 6525 ED Nijmegen, The Netherlands. [3]Los Alamos National Laboratory, Los Alamos, NM 87545, USA. [4]Department of Physics and Astronomy, Rice Center for Quantum Materials, Rice University, Houston, TX 77005, USA. ✉e-mail: paschen@ifp.tuwien.ac.at

(i) an electronic specific heat coefficient $\Delta C/T$ that is linear in $T^2$, with a giant slope, evidencing ultraslow quasiparticles with linear electronic dispersion[12]; and (ii) a giant spontaneous Hall effect as a result of Weyl nodes—sources and sinks of Berry curvature—pinned to the immediate vicinity of the Fermi level[11]. We note that the semimetallic ground state of $Ce_3Bi_4Pd_3$ is well documented (Supplementary Note 2). As will be shown in what follows, the momentum-space separation of the Weyl nodes, which are positioned in a Kondo insulating background, is successively reduced with increasing magnetic field, until the Weyl nodes meet in momentum space and annihilate. Only at considerably larger magnetic fields, the Kondo insulator gap is quenched and the system becomes a heavy fermion metal (see Supplementary Note 3 and Supplementary Fig. 2 for a cartoon of the correlated electronic bandstructure). The latter transition has also been observed in[14].

Figure 1 gives an overview of our electrical transport and specific heat data. Salient transport features in zero magnetic field are an electrical resistivity that increases moderately with decreasing temperature (Fig. 1a) and a linear-response Hall coefficient with two ranges of thermally activated behavior and a saturation to a constant value at the lowest temperatures (Fig. 1b). A plausible interpretation, which will receive further support from the field-dependent data presented below, is that this behavior results from a (pseudo)gapped background density of states, within which a narrow (Kondo insulator) gap forms at lower temperatures, against which a small residual density of states associated with the Fermi-level-bound Weyl nodes becomes apparent

at the lowest temperatures (see cartoon in Supplementary Fig. 2). The presence of a (pseudo)gap in the noninteracting density of states is in agreement with density functional theory (DFT) calculations, although the theoretical gap size is considerably larger[15].

The application of magnetic fields gradually suppresses the low-temperature resistivity upturn. This is the case until, ultimately, metallic behavior is seen, albeit with higher resistivity and a different temperature dependence than in the nonmagnetic reference compound $La_3Bi_4Pd_3$ (Fig. 1a). This indicates that Kondo physics is at play even at our largest field of 37 T. Isothermal magnetic-field-dependent measurements of the electrical resistivity $\rho_{xx}(B)$ (Fig. 1c) and Hall resistivity $\rho_{xy}(B)$ (Fig. 1d) reveal that this field-induced transformation occurs in two stages. The signatures thereof are most pronounced in the lowest-temperature data. Here, the electrical resistivity displays a shoulder at about 9 T and a crossover to almost field-independent behavior at about 14 T (see arrows labeled $B_{c1}$ and $B_{c2}$, respectively, and Supplementary Note 4 for further analyses). The corresponding signatures in the Hall resistivity are kinks at the same two fields. A quantitative analysis, presented further below (Fig. 2), reveals that this behavior reflects a two-stage Fermi surface reconstruction at two quantum phase transitions.

We first examine the effect the magnetic field has on the material's topological characteristics. The most direct signature is a giant spontaneous as well as even-in-field nonlinear topological Hall effect, which evidences Berry curvature singularities from Weyl nodes in close

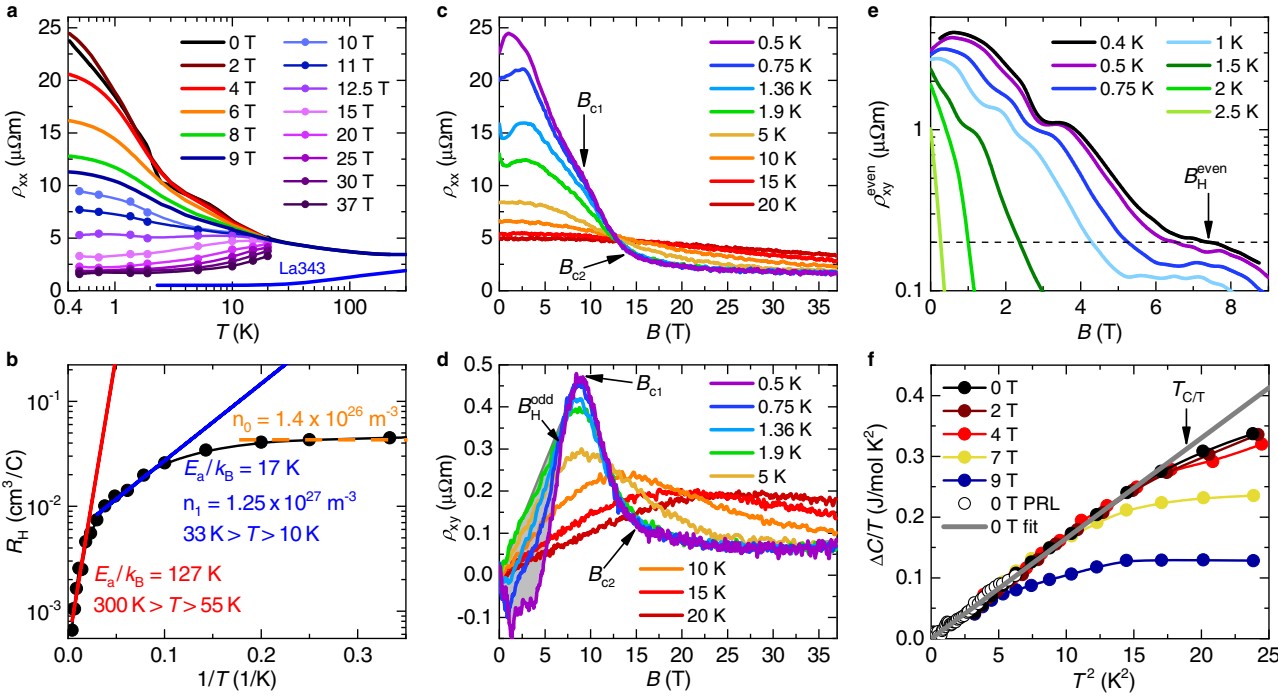

**Fig. 1 | Data overview and characteristic transport and thermodynamic scales of $Ce_3Bi_4Pd_3$. a** Temperature-dependent electrical resistivity at various fixed magnetic fields. For $B > 9$ T, iso-$B$ cuts (dots) were taken from panel (**c**). The zero-field resistivity of the nonmagnetic reference compound $La_3Bi_4Pd_3$ is shown for comparison. The small kink seen near 2.5 K in the zero-field data is associated with the onset of a giant spontaneous Hall voltage that, due to the large associated Hall angle, leaves an imprint also on the longitudinal resistivity[11]. The low-field data are taken from Ref. 11. **b** Arrhenius plot of the linear response normal (antisymmetrized) Hall coefficient $R_H$ of $Ce_3Bi_4Pd_3$ (black symbols). The black line is a guide to the eyes. Red and blue lines correspond to fits with $R_H = R_{H,i}\exp[E_a/(k_B T)]$, where $n_i = 1/(R_{H,i}e)$ is the charge carrier concentration in a simple one-band model. Below 10 K, the data saturate to a constant value (dashed orange line), with the charge carrier concentration $n_0$ (again in a one-band model). **c** Magnetoresistance isotherms in fields up to $B = 37$ T, for various temperatures between 0.5 and 20 K.

**d** Normal (antisymmetrized) Hall resistivity isotherms at the same fields and temperatures. $B_{c1}$ and $B_{c2}$ are determined in Fig. 2. Below $B_H^{odd}$ (shown for the 0.5 K data), the anomalous Hall contribution (grey shading) leads to a deviation from the initial linear-in-$B$ normal Hall resistivity by more than 5%. **e** Even-in-field (symmetrized, see Supplementary Note 6) Hall resistivity isotherms at low temperatures and fields. Above $B_H^{even}$ (shown for the 0.4 K data) this Weyl-node derived signal drops to below 0.2, which is 5% of the maximum of the lowest-temperature isotherm (data from Ref. 11). **f** Temperature-dependent electronic specific heat coefficient (see Supplementary Note 5 for details) vs $T^2$ at various fixed fields (open symbols are from Ref. 12). Above $T_{C/T}$ (shown for the 0 T data), the data deviate by more than 5% from the low-temperature $\Delta C/T = \Gamma T^2$ fit, which represents the linear Weyl dispersion. The spurious low-field features in the low-temperature resistivity isotherms (Fig. 1c) are imprints of the large Hall contributions, caused by the current path redistribution due to the large Hall angle[11].

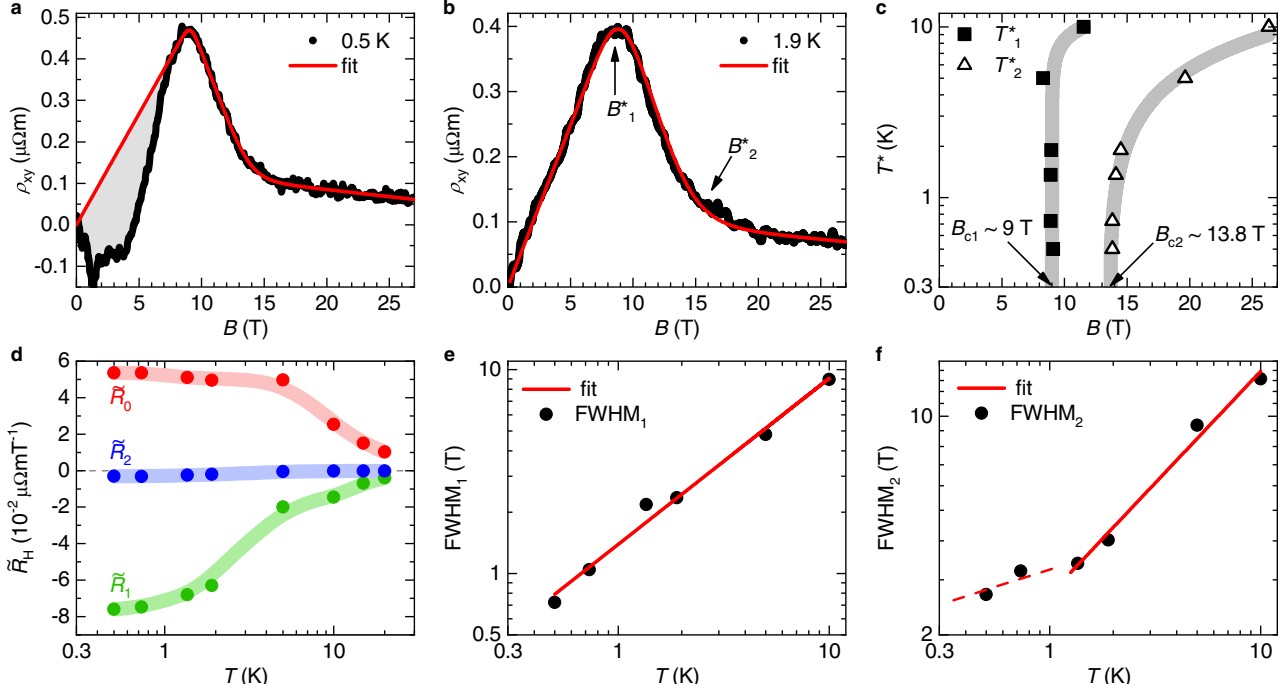

**Fig. 2 | Two-stage Fermi surface reconstruction in Ce$_3$Bi$_4$Pd$_3$. a** Hall resistivity vs applied magnetic field at 0.5 K, together with the best fit according to the two-crossovers model (see Supplementary Note 7). The deviation of the data from linear behavior (shaded grey area) is due to a Berry curvature-derived anomalous Hall contribution (Ref. [11]). **b** Same as (a) at 1.9 K. The Berry-curvature derived contribution is essentially absent at this temperature. The two characteristic fields $B_1^*$ and $B_2^*$ mark the positions of the crossovers between the different regimes. **c** The crossover fields $B_1^*(T)$ and $B_2^*(T)$ determined from the $\rho_{xy}(B)$ fits are plotted as $T_1^*(B)$ and $T_2^*(B)$, respectively, in a temperature-magnetic field phase diagram. They extrapolate, in the zero-temperature limit, to $B_{c1}$ and $B_{c2}$ and separate three regimes of simple (linear-in-$B$) normal Hall resistivity. **d** Differential Hall coefficients $\tilde{R}_0$, $\tilde{R}_1$, and $\tilde{R}_2$ of these three regimes, determined from the $\rho_{xy}(B)$ fits, as function of temperature. The lines are guides to the eyes. **e** Width of the first crossover FWHM$_1$, representing the full width at half maximum of the second derivative of the two-crossovers fit function. The straight line is a pure power law fit, FWHM$_1 \propto T^p$, with $p = 0.81$. It describes the data down to the lowest temperature, evidencing that the carrier concentration changes abruptly at $T = 0$. **f** Same as in (e) for the second crossover. The straight full line is a pure power law fit, FWHM$_2 \propto T^p$, with $p = 0.71$ to the data above 1 K. At lower temperatures, the rate of decrease is somewhat reduced (red dashed line).

vicinity to the Fermi energy[11]. Somewhat more indirect evidence is a temperature-dependent electronic specific heat that varies linearly in $T^3$, with a slope that even surpasses the (Debye-like) phonon contribution, and evidences linearly-dispersing electronic bands with ultralow velocity[12,13]. Together they have established the inversion symmetry (IS)-broken (noncentrosymmetric and nonsymmorphic) but time reversal symmetry (TRS)-preserving heavy fermion compound Ce$_3$Bi$_4$Pd$_3$ as a model case of a strongly correlated topological semimetal[11,12] (see Supplementary Note 1 for further details).

In Fig. 1e we show how isothermal even-in-field (symmetrized and corrected for contact misalignment, see Supplementary Note 6 with Supplementary Fig. 5, and Ref. [11]) topological Hall resistivities $\rho_{xy}^{even}$ are successively suppressed by magnetic field. The apparent fine structure in this suppression, seen in the isotherms below 2 K, may reflect various regimes of Weyl node configurations in momentum space. Indeed, a rich sequence of Weyl node motion and annihilation under magnetic field tuning was found in Kondo model calculations on a diamond lattice[16,17], which is an interesting topic for further investigations. Here, we focus on the ultimate total suppression of the effect, which occurs at $B_H^{even}$ (see arrow in Fig. 1e for the lowest temperature isotherm). Also the linear-in-$T^3$ electronic specific heat (corresponding to an electronic specific heat coefficient $\Delta C/T = \Gamma T^2$, see Supplementary Note 5 for details) is successively suppressed by magnetic fields, which we quantify by the parameter $T_{C/T}$ (see arrow in Fig. 1f on the zero-field curve). Interestingly, this suppression happens at constant $\Gamma$, indicating that the shape (slope and energy) of the Weyl dispersion remains unchanged as the Weyl nodes move in momentum space. Finally, we point to another feature that accompanies these two key signatures. It appears as an anomaly in the (normal, antisymmetrized and thus

odd-in-field) Hall resistivity isotherms (see arrow denoting $B_H^{odd}$ as upper end of the grey shading, marked on the lowest-temperature isotherm in Fig. 1d) and is known as the anomalous topological Hall effect in TRS-broken Weyl semimetals[18]. It is associated with a magnetic field-induced even-in-momentum Berry curvature, which is in addition to the intrinsic (zero-field) odd-in-momentum Berry curvature of Ce$_3$Bi$_4$Pd$_3$ (see also Ref. [11]).

All these results together establish that magnetic field quenches the topological response, apparently via a process that moves the Weyl nodes at equal energy in momentum space. This raises the following questions: Which mechanism underlies this effect? Does magnetic field suppress the Kondo effect just as increasing the temperature above the Kondo coherence scale does[11], thereby removing the correlated electrons and thus the basis for the Weyl-Kondo semimetal formation? Or did we succeed to annihilate the Weyl nodes in an intact Kondo coherent system? The Kondo-like electrical resistivity in 37 T suggests that some form of Kondo correlations persist. To show that the magnetic field indeed controls Kondo-driven Weyl nodal excitations, however, what needs to be established is that the Kondo effect as realized in zero magnetic field operates over the entire field range with topological response. We now turn to the search for such evidence.

We start with a quantitative analysis of the (normal, antisymmetrized) Hall resistivity isotherms $\rho_{xy}(B)$ of Fig. 1d. As established previously[7], when magnetic field drives transitions between ground states with different Fermi volumes, the resulting (finite temperature) crossovers manifest as (broadened) kinks in $\rho_{xy}(B)$, and (broadened) steps in the differential Hall coefficient $\tilde{R}_H(B) = \partial\rho_{xy}(B)/\partial B$. Such behavior has been observed in a number of heavy fermion metals[7,19–21] driven by magnetic field across quantum critical points. From fits with

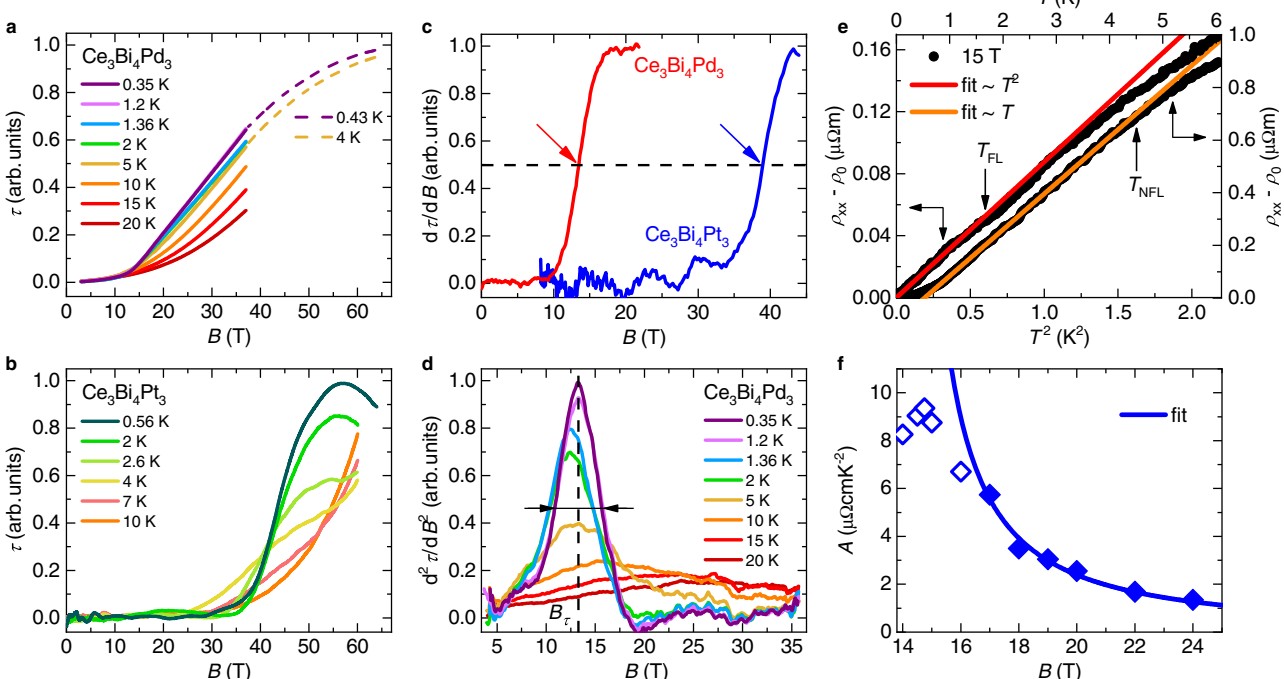

**Fig. 3 | Field-tuned Kondo insulator to heavy fermion metal transition in Ce₃Bi₄Pd₃. a** Magnetic torque isotherms of $Ce_3Bi_4Pd_3$ at various temperatures as function of applied magnetic field up to 37 T and, for low temperatures, up to 65 T. **b** Same as in (**a**) for the Kondo insulator $Ce_3Bi_4Pt_3$. **c** Field derivative of the magnetic torque isotherms of $Ce_3Bi_4Pd_3$ and $Ce_3Bi_4Pt_3$ at the lowest temperatures, revealing similar characteristics, albeit at different fields (the fields where the step-like increases reach half height are indicated by arrows). **d** Second field derivative of the torque signal of $Ce_3Bi_4Pd_3$. We define the characteristic field $B_\tau$ of the torque signal as the center of the width at half maximum, as indicated by the dashed vertical line

and the horizontal arrows for the 0.35 K isotherm. **e** Electrical resistivity of $Ce_3Bi_4Pd_3$ at 15 T, displaying linear-in-$T^2$ Fermi liquid behavior from the lowest temperature of 0.1 K up to $T_{FL}$ (bottom and left axes, red line; above $T_{FL}$, the data deviate by more than 5% from the Fermi liquid form $\rho = \rho_0 + AT^2$) and linear-in-$T$ non-Fermi liquid behavior from somewhat above $T_{FL}$ up to $T_{NFL}$ (top and right axes, orange line; see Supplementary Note 11 for details). **f** Fermi liquid $A$ coefficient vs applied magnetic field. The data above 16 T are well described by $A \propto (B - B_{c2})^{-p}$ with $B_{c2} = 13.8$ T and $p \approx 1$. Deviations at lower fields are attributed to proximity to the Kondo insulating phase (see Supplementary Note 11 and Supplementary Fig. 11).

a phenomenological crossover function[7] (see Supplementary Note 7 for details) one can extract not only the $\tilde{R}_H$ values associated with the different phases, but also the crossover fields $B^*$ and sharpnesses, quantified by the full width at half maximum (FWHM). In Fig. 2a, b we show two representative fits, for data at 0.5 K (for which we have subtracted the above-discussed anomalous topological Hall effect contribution) and 1.9 K, respectively. Fits of similar quality are obtained at all temperatures up to 10 K (Supplementary Fig. 6). At higher temperatures, we lose track of the two-stage nature and thus this model does no longer give meaningful results. Note that anomalous (nontopological) Hall contributions and multiband effects do not play significant roles here (see Supplementary Notes 8 and 9). The temperature-dependent fit parameters are shown in Fig. 2c–f. The two crossover fields $B_1^*$ and $B_2^*$ (Fig. 2b), determined for all available isotherms, are plotted as characteristic temperatures $T_1^*(B)$ and $T_2^*(B)$ in Fig. 2c. The extrapolations to $T = 0$ of these curves identify $B_{c1}$ and $B_{c2}$ (see arrows in Fig. 2c). Both crossovers sharpen considerably with decreasing temperature (Fig. 2e, f), indicating that the phase diagram of magnetic field-tuned $Ce_3Bi_4Pd_3$ comprises three phases with distinct Fermi volumes: a phase below $B_{c1}$ with a small hole-like Fermi volume, an intermediate-field phase between $B_{c1}$ and $B_{c2}$ with an even smaller electron-like Fermi volume, and a high-field phase beyond $B_{c2}$ with a much larger Fermi volume.

To understand this behavior and elucidate the character of these phases we have carried out torque magnetometry measurements. At low magnetic fields, and in particular across $B_1^*(T)$, no sizable torque signal is detected (Fig. 3a), thus ruling out that a magnetic phase transition occurs at this field (see Supplementary Note 10 and Supplementary Fig. 10). A pronounced torque signal appears only above about 14 T; it corresponds to the onset of nonlinearity in the

magnetization[14]. Similar behavior, albeit with a much larger magnetic field scale, is also seen in the canonical Kondo insulator $Ce_3Bi_4Pt_3$, which we have studied for comparison (Fig. 3b). The corresponding characteristics in the first and second derivative with respect to the magnetic field are a step-like increase and a maximum, respectively. For $Ce_3Bi_4Pt_3$, the step in the lowest temperature isotherm occurs at 38.9 T (blue arrow in Fig. 3c), which is close to the field where a Kondo insulator to metal transition has previously been evidenced by a jump of the Sommerfeld coefficient[22]. The very similar feature seen for $Ce_3Bi_4Pd_3$ (red curve in Fig. 3c) then suggests that also in this system a Kondo insulator to metal transition takes place, albeit at much lower fields. As characteristic field $B_\tau$ of this transition, which is well-defined for all isotherms, we use the middle field at half height of the second derivative curves (Fig. 3d).

We also performed temperature-dependent electrical resistivity measurements at fields around this Kondo insulator to metal transition. On the high-field side of the transition, we observe Fermi liquid behavior, $\rho = \rho_0 + AT^2$ (see Fig. 3e, bottom and left axes, for data at 15 T). This confirms that $Ce_3Bi_4Pd_3$ has indeed metallized. The $A$ coefficient measures the strength of electronic correlations. Values in the range of several $\mu\Omega cm/K^2$, as observed here, are typical of heavy fermion metals[23] (see Supplementary Note 11 for details). Thus, the quenching of the Kondo insulator gap by the magnetic field has indeed (as already indicated by the high-field resistivity curves in Fig. 1a) still not suppressed the Kondo interaction. Thus, what happens at $B_{c2}$ is a field-induced Kondo insulator to heavy fermion metal transition, which was also studied in Ref. 14. In Fig. 3f we plot the $A$ coefficient determined also for other fields (see Supplementary Fig. 11) as function of the magnetic field. Upon approaching the transition from the high-field side, the $A$ coefficient increases, and is well described by a

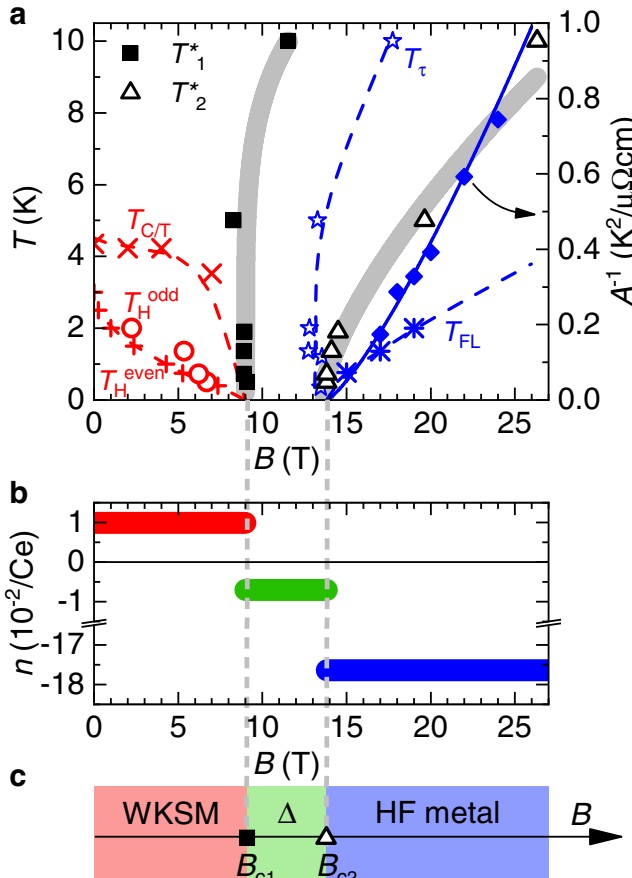

**Fig. 4 | Magnetic-field tuned phase diagram of $Ce_3Bi_4Pd_3$. a** Temperature-magnetic field phase diagram with the crossover temperatures $T_1^*$ and $T_2^*$ (from Fig. 2c), the temperature scales $T_{C/T}$, $T_H^{even}$, and $T_H^{odd}$ (all from Fig. 1) associated with the Weyl-Kondo semimetal phase, the characteristic scales $T_\tau$ of the torque signal and $T_{FL}$ of Fermi liquid behavior (both from Fig. 3) (all left axis). The inverse $A$ coefficient (from Fig. 3) is plotted on the right axis, indicating an effective mass divergence at $B_{c2}$. **b** Effective charge carrier concentration in a single-band model, $n = 1/(\tilde{R}e)$, at the lowest temperature of 0.5 K (from Fig. 2d), in the three different magnetic field ranges. **c** Axis across a theoretical zero-temperature phase diagram. A topological quantum phase transition between a Weyl-Kondo semimetal (WKSM) and a phase with annihilated and thus gapped-out Weyl nodes (Δ) occurs at $B_{c1}$. The underlying Kondo insulator gap persists across both the WKSM and the Δ phase, and is quenched only at a quantum critical point at $B_{c2}$ to reach a heavy fermion (HF) metal phase at high fields.

extending those measurements down to the temperature range of dilution refrigerators). Clearly, they do not represent normal (metallic) behavior and thus we define the upper boundary of Fermi liquid behavior $T_{FL}$ (see arrows in Fig. 3e and Supplementary Fig. 11) only in the field range where these effects have minor influence. We conclude that a heavy Fermi liquid phase exists at fields above $B_{c2}$. In turn, this implies that the Kondo effect as operating at $B = 0$ remains intact across $B_{c1}$.

We are now in the position to construct a temperature-magnetic field phase diagram that assembles the above-discussed characteristics (Fig. 4a). At low fields, these are $T_{C/T}(B)$, $T_H^{even}(B)$, and $T_H^{odd}(B)$, all denoting temperatures up to which, at a given field, a certain signature of Weyl-Kondo semimetal behavior is detected. These scales all collapse at a critical field $B_{c1}$ of 9 T. Importantly, the correlated background remains essentially unchanged across this field and no magnetic phase transition takes place. At high fields, we plot the scales $T_\tau(B)$ and $T_{FL}(B)$ associated with the torque anomaly and Fermi liquid behavior, respectively, and also include the inverse of the Fermi liquid resistivity coefficient, $1/A$, which hits zero at the effective mass divergence. In addition, we show the $T_1^*(B)$ and $T_2^*(B)$ scales extracted from our quantitative Hall resistivity analysis. Note that all these temperature scales are crossovers; they do not represent phase boundaries in the thermodynamic sense as none of the phases is associated with an order parameter.

In Fig. 4b we plot the charge carrier concentration $n$ (in units of carriers per Ce atom) of the three phases, extracted in a single-band model from the lowest-temperature values of the differential Hall coefficients (Fig. 2d). It changes from about 0.01 holes below $B_{c1}$, via 0.007 electrons between $B_{c1}$ and $B_{c2}$, to 0.18 electrons beyond $B_{c2}$. The 25-fold increase across $B_{c2}$ is independent evidence for the above-discussed Kondo insulator to heavy fermion metal transition. The more modest reduction of the absolute value of the carrier concentration across $B_{c1}$ indicates a Fermi surface reconstruction that involves only a fraction of momentum space. The annihilation and associated gapping-out of Weyl nodes is exactly such a phenomenon. Also the sign change of $n$ across $B_{c1}$ can be understood in this scenario (see Supplementary Note 12 and Supplementary Fig. 12). That $n$ is field independent below $B_{c1}$ (as seen from the linear-in-field behavior of the Hall resistivity in this regime, Fig. 2b) confirms that magnetic field moves the Weyl nodes at constant energy. The only process that can then remove them is their mutual annihilation, which we thus propose to happen at $B_{c1}$. Importantly, this takes place in a background of unchanged symmetry and with the Kondo effect continuing to operate. In other words, we have realized a controlled suppression of the Kondo-driven topological semimetal. A corollary is that we have isolated the tuning of topology from the tuning of correlation physics as such.

In Fig. 4c we summarize these findings in a schematic zero-temperature phase diagram. The low-field phase is a Weyl-Kondo semimetal (WKSM)[11–13] that, as evidenced here, consists of Weyl nodes situated within the narrow energy gap of a Kondo insulator. As function of magnetic field a sequence of two Fermi volume-changing quantum phase transitions is observed. At $B_{c1}$, all signatures of the Weyl-Kondo semimetal disappear as the Weyl nodes annihilate in a topological quantum phase transition. At $B_{c2}$, the Kondo insulator with gapped-out Weyl nodes (denoted by the symbol Δ) transforms into a heavy fermion (HF) metal, in a transition that displays signatures of quantum criticality and must thus be at least nearly continuous.

This raises the question of why the phenomena of genuine topology tuning and Weyl node annihilation have remained elusive in the much more extensively studied noninteracting regime[29]? A magnetic field can either act on the spin (via the Zeeman effect) or the charge of an electron (via the orbital effect). In Kondo systems the former dominates, in high-mobility semimetals the latter. The momentum space motion of Weyl nodes requires sizable Zeeman coupling[16,17]. In its absence[30], field-driven changes of topological

divergence, $A \propto 1/(B - B_{c2})$, with the critical field $B_{c2} = 13.8$ T from the Hall resistivity analysis (see caption of Fig. 3 for details). Such behavior is known from heavy fermion metals tuned by a magnetic field to a quantum critical point (QCP)[21,24]. In this case, also non-Fermi liquid (NFL) behavior should develop[25,26], which we indeed observe in the form of a linear-in-temperature resistivity at 15 T, at temperatures above the Fermi liquid behavior seen at the lowest temperatures (Fig. 3e, top and right axes). This confirms that, at 15 T, $Ce_3Bi_4Pd_3$ is slightly away from a QCP. Closer to the expected quantum critical field $B_{c2}$, the resistivity appears to be influenced by the nearby Kondo insulator phase, as seen by a rapid suppression of the $A$ coefficient and a crossover to $T^2$ behavior with a negative slope (Supplementary Fig. 11h), as well as an increase of the residual resistivity $\rho_0$ not only towards but even across $B_{c2}$ (Supplementary Fig. 11i). Whether these (nonmetallic) characteristics are generic to field-induced Kondo insulator to heavy fermion metal transitions is an interesting topic for future studies (we note that in a related pressure-induced transition in $SmB_6$ no such effects were seen[27,28], but this may reflect the need of

signatures[31,32] can arise from orbital effects such as the tunneling between zeroth Landau level states of adjacent Weyl nodes[31], at finite Weyl node separation. Of course, also changes in a material's broken symmetry state can be accompanied by changes of topology[33], but this is not the topic of interest to us here (see Supplementary Notes 13 and 14 for further details).

In summary, we have demonstrated the genuine control of Weyl nodes, with clarity and ease. The clarity is attributed to the fact that in $Ce_3Bi_4Pd_3$ the Weyl nodes form within a (strongly correlated, Kondo insulating) gapped state and are positioned in close vicinity of the Fermi energy. Because topologically trivial states are gapped out, there is no need to disentangle Weyl fermions from topologically trivial carriers, which hampers the field of weakly interacting Weyl semimetals. The ease of control—namely that a rather modest field of 9 T was not only enough to manipulate the positions of the Weyl nodes in momentum space, but even drive their annihilation—shows that the well-known excellent tunability of (topologically trivial) strongly correlated electron systems[1–4] holds also for topological features in such systems. As such, our study lays the ground for establishing a global phase diagram for strongly correlated topological materials. Key open questions to address include whether the phases and transitions discovered here exist also in other materials and are thus universal, and whether quantum criticality plays an important role in stabilizing them. The latter is hinted at by recent inelastic neutron scattering experiments[34].

Our findings may also guide investigations in related materials classes. Much effort is currently devoted to artificial materials such as twisted bilayer systems where correlations can be enhanced via a moiré potential[35–38]. Additional tuning knobs in such heterostructures are the dielectric displacement and electrostatic doping, which may become powerful if further advances towards highly reproducible structures can be accomplished. Finally, we point to the potential of strongly correlated bulk materials such as $Ce_3Bi_4Pd_3$ for quantum devices[4,39–41], where the robust and giant topological response—together with the high level of topology control demonstrated here—opens new opportunities. As an example we name microwave nonreciprocity at zero magnetic field, a key functionality needed in circuit quantum electrodynamics systems[42], that could be realized via the spontaneous Hall response of $Ce_3Bi_4Pd_3$.

## Methods

### Synthesis

Single crystals of $Ce_3Bi_4Pd_3$, $Ce_3Bi_4Pt_3$, and of the nonmagnetic reference compound $La_3Bi_4Pd_3$ were grown using the flux method. For $Ce_3Bi_4Pt_3$, elementary Ce, Pt, and Bi in an atomic ratio of 1:1:7 were placed in an alumina crucible, and heated to 1100°C in a vacuum-sealed quartz tube, using a box furnace. The melt was then slowly cooled to 600°C after a dwell time of 12 h at 1100°C, with a cooling rate of 1°C/h, and then left annealing for 12 h. Crystals of typically 1 mm in diameter were then extracted from the melt using a centrifuge. For $Ce_3Bi_4Pd_3$ and $La_3Bi_4Pd_3$, as the primary stable phases at Bi excess are $CeBi_2Pd$ and $LaBi_2Pd$, the Bi content was strongly reduced to about 1:1:1.5 (see also Ref. 12). The chemical composition and crystal structure of the samples were determined by energy dispersive x-ray spectroscopy and powder x-ray diffraction. Laue diffraction was utilized to determine the crystallographic orientation of selected samples.

### Measurement setups

**High-field experiments.** Magnetoresistance, Hall effect, and torque magnetization measurements in DC fields up to 37 T were performed at the HFML-EMFL facility at Nijmegen. Magnetotransport data were measured using Stanford Research SR830 lock-in amplifiers, with the measured voltage signal pre-amplified 100 times using Princeton Applied Research low-noise transformers. Electrical contacts where made by either spot welding or gluing with silver paint 12 $\mu$m diameter gold wires to the samples in a 5-wire configuration. All displayed Hall resistivity curves were obtained by the standard antisymmetrizing procedure of the resistivity $\rho_{xy}^{meas}$ measured across the Hall contacts, i.e., $\rho_{xy}(B) = [\rho_{xy}^{meas}(+B) - \rho_{xy}^{meas}(-B)]/2$. This cancels out both the spontaneous Hall effect (zero-field signal) and any even-in-field component. Torque and magnetization measurements in pulsed fields up to 65 T were performed at the NHMFL-LANL facility at Los Alamos. Magnetization data were obtained only for $Ce_3Bi_4Pt_3$, using an extraction magnetometer and the "sample-in/sample-out" technique to separate the sample signal from the background. In this case, the magnetic field was not aligned to any particular crystallographic axis. In all torque experiments, piezo-cantilevers were used for enhanced sensitivity. Samples were attached to the levers with Dow Corning high vacuum grease. The measured torque signal was obtained after balancing a Wheatstone bridge containing the resistance of the sample's cantilever and a "dummy" (empty lever) resistor. Due to the small samples required for this technique, the magnetic field was initially not aligned to any particular crystallographic axis. A sample rotator was used to scan the torque signal across different orientations. In all cases, temperatures down to 0.35 K were obtained using a $^3$He cryostat.

**Low-temperature experiments.** Additional magnetoresistance, Hall effect, and specific heat measurements were obtained in Vienna using a Quantum Design Physical Property Measurement System, equipped with $^3$He options. Low-temperature electrical resistivity measurements down to 70 mK and in magnetic fields up to 15 T were performed in an Oxford dilution refrigerator.

## Data availability

All data that are necessary to interpret, verify, and extend the presented research are contained in the main part of this article. They are provided through deposition in the repository Zenodo (https://doi.org/10.5281/zenodo.7043820).

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

## Acknowledgements

The authors wish to thank H.-H. Lai, S. E. Grefe, and A. P. Higginbotham for fruitful discussions. We acknowledge support of the HFML-RU/NWO-I, member of the European Magnetic Field Laboratory (EMFL). A portion of this work was performed at the National High Magnetic Field Laboratory, which is supported by the National Science Foundation Cooperative Agreement No. DMR-1157490 and DMR-1644779, the State of Florida and the United States Department of Energy. M.J. acknowledges support from the US DOE Basic Energy Science project "Science at 100T". The work in Vienna was supported by the Austrian Science Fund (I2535, S.P.; I4047, D.Z.; 29279, S.P.; I5868–FOR5249, S.P.), the European Union's Horizon 2020 Research and Innovation Programme (824109-EMP, S.P.), and the European Research Council (ERC Advanced Grant 101055088-CorMeTop, S.P.). The work at Rice was in part supported by the NSF (DMR-2220603, Q.S.), the AFOSR (FA9550-21-1-0356, Q.S.), and the Robert A. Welch Foundation (C-1411, Q.S.).

## Author contributions

S.P. designed and guided the research. X.Y. and A.P. synthesized and characterized the material. S.D., D.Z., A.M., F.W., R.M., L.T., B.V., L.E.W., and M.J. performed the high-field experiments, M.T., S.D., G.E., and D.Z. the low-temperature experiments. S.D. analyzed the data, with contributions from D.Z., M.T., G.E., and S.P. The manuscript was written by S.P., with contributions from S.D., D.Z., and Q.S. All authors contributed to the discussion.

## Competing interests

The authors declare no competing interests.
