## [Peer Review File · Nature Communications]

REVIEWER COMMENTS

Reviewer #1 (Remarks to the Author):

Referee report “Controlling electronic topology in a strongly correlated electron systems” by Dzaber et al.

The authors report the results of transport and thermodynamic measurements in $\text{Ce}_3\text{Bi}_4\text{Pd}_3$ in magnetic fields and low temperatures. The authors interpret their results by understanding the observed features in magneto-resistivity as a Zeeman-driven motion of Weyl nodes in momentum space. This manuscript appears to be third in a series of three papers: the first one [published in PRL] discussed the isovalent substitutions of Pt with Pd in $\text{Ce}_3\text{Bi}_4\text{Pt}_{1-x}\text{Pd}_x$, which lead to the emergence of the semimetallic response in heat capacity with increase in x , while the second paper [PNAS] reported to the observation of the giant Hall response, which the authors - as far as I could understand - attributed to the existence of the Berry dipole in the band structure of this material.

Below I list my comments first:

1. I think the novelty of the results – when taken in general context - are questionable. For example, the title of the manuscript directly refers to the topic which has been already actively discussed during the last few years in the context of SmB_6 . Indeed, according to the second sentence in the second paragraph, SmB_6 can be classified as a strongly correlated material, which under the application of external pressure exhibits a crossover from semiconducting-like to metallic-like in resistivity. This is an example of how one can control electronic topology in strongly correlated systems. The authors are undoubtedly aware of the ongoing debate on whether SmB_6 hosts topologically protected metallic surface states and that (i) there is convincing experimental evidence that this is indeed the case and (ii) numerous attempts to probe the evolution of the surface states under the influence of disorder, magnetic field and/or pressure. That is why I would like to know why the authors do not cite any papers on SmB_6 which are clearly relevant to the subject of this paper [they have done it in their PRL paper]. Consequently, I suggest that the authors either provide the relevant citations already in the introductory part of their manuscript or change the title of their manuscript which would make analogies with the physics of samarium hexaboride much less obvious.

2. As I am reading through the manuscript, I encounter the words like “... a plausible interpretation”, “may reflect ...” and “... likely via...”, which as a reader I interpret as if the authors are not sure what their experimental results really mean and, so after all, my impression is that author’s interpretation

may well be correct, but it also may well be wrong. I am afraid this is not good enough: this is an experimental paper and the experimental data should be good enough in order to avoid being overinterpreted in one way or another.

3. On page 2, in the third paragraph the authors claim that they discover the “topological quantum phase transition.” As we know, quantum phase transitions are usually described by the correlation functions, so what correlation function could be probed here? If the authors mean the trivial transition between the states with two different topological indices, than I suggest that they say so explicitly.

4. In the same paragraph, the last sentence the authors claim that their discovery ‘opens up entirely new regime in the correlation topology interplay’. Again, in my opinion, this is a purely declarative statement. What do you mean by that?! Are you discovering here something fundamentally new and, so that, say, transitions in SmB6 induced by external pressure and/or magnetic field do not fall under this ‘regime of correlation topology interplay’? Please, be more specific and straightforward.

5. The paper severely lacks the numerical estimates so that the reader could at least get some idea in which regime the system is. For example, I think it would be helpful if the authors could explain whether one should consider a quasi-classical limit [when magnitude of magnetic field is much less than the effective Fermi energy] or a quantum one?

Unfortunately, I cannot recommend this manuscript for publication in the present form.

Reviewer #2 (Remarks to the Author):

In this work the authors study Ce₃Bi₄Pd₃ with magneto transport and torque magnetometry measurements. Their data indicate that the Weyl nodes in this system respond to magnetic fields by moving in momentum space and ultimately annihilating. This is an interesting and well written paper and certainly worthy of publication in Nature communications. However I have a few comments that I would like the authors to address before recommending publication.

1) There is uncertainty about the actual ground state of Ce₃Bi₄Pd₃. For example, a recent work (arXiv:2110.08230v1) suggests it is a narrow-gap Kondo insulator. I would urge the authors to start with a short summary of what is known about this materials class, and describe the two competing scenarios. This should then be revisited in the discussion.

2) The phenomenology described here is complex. A schematic of the band structure of Ce₃Bi₄Pd₃ at a couple of important temperatures (at zero field) and the authors conjecture for how the band structure changes with field (again at a couple of important temperatures) would be very useful.

Reviewer #3 (Remarks to the Author):

The authors report a topological quantum phase transition of a strongly correlated Weyl-Kondo semimetal Ce₃Pd₄Bi₃ in strong magnetic fields. The phase transition contains two stages, from a low-field Weyl-Kondo semimetal with small hole-like fermi volume to an intermediate-field Kondo insulator with small electron-like fermi volume, and then to a high-field heavy fermion metal with large fermi volume. The transition from the Weyl-Kondo semimetal to Kondo insulator is topological because the even part of the Hall signal vanishes at 9T and the linear-in-T³ electronic specific heat is suppressed by magnetic fields. The authors conclude this transition as field-driven, Weyl nodes annihilation. More evidences in Hall and magnetic signals support the existence of the transition from a Kondo insulator to heavy fermion metal.

This draft contains sufficient and interesting data and I would like to support it to be published in Nature communications. Yet I suggest the authors to consider a serious revise because the current draft is difficult to follow if the readers did not read through ref. 6 and 7. Below I have some suggestions.

1. Figure 1 shows some notation like ρ^{even} . Consider the readers cannot understand it if they did not read Ref 7. I suggest the authors to add a panel of data analysis in this figure like Fig. 3A in ref. 7.

2. I suggest the authors to rewrite the introduction part. The first paragraph in page 3 is not helpful yet there is lack of introduction of Ce₃Pd₄Bi₃. The authors may consider move some sentences from SI part II to the introduction part.

3. The spontaneous Hall effect is important for supporting the existence of Weyl fermion because it directly demonstrates the Berry curvature field. Why the intermediate-field Kondo insulator with small electron pockets has no Berry curvature field?

Reply to the Reviewers

We thank all three Reviewers for taking the time to review our manuscript, for their overall positive assessments, and their thoughtful comments. All three seem to see the potential of our work and propose some revisions. We believe that we could fully address all their points, and that our revisions have further improved the quality of the manuscript. We thus hope that the Reviewers now find our work suitable for publication in Nature Communications.

For clarity, we quote the Reviewers' comments in *italic* and type our replies as normal text. Changes to the manuscript and Supplementary Information are indicated after each point. In addition, a copy of the revised manuscript with all changes marked in red is provided. Note that line, figure, section, and reference numbers refer to the revised manuscript.

Reviewer #1

Referee report "Controlling electronic topology in a strongly correlated electron systems" by Dzaber et al.

The authors report the results of transport and thermodynamic measurements in $Ce_3Bi_4Pd_3$ in magnetic fields and low temperatures. The authors interpret their results by understanding the observed features in magneto-resistivity as a Zeeman-driven motion of Weyl nodes in momentum space. This manuscript appears to be third in a series of three papers: the first one [published in PRL] discussed the isovalent substitutions of Pt with Pd in $Ce_3Bi_4Pt_{1-x}Pd_x$, which lead to the emergence of the semimetallic response in heat capacity with increase in x , while the second paper [PNAS] reported to the observation of the giant Hall response, which the authors - as far as I could understand - attributed to the existence of the Berry dipole in the band structure of this material.

We thank the Reviewer for his/her concise summary of our previous achievements.

Below I list my comments first:

- 1.1 *I think the novelty of the results – when taken in general context – are questionable. For example, the title of the manuscript directly refers to the topic which has been already actively discussed during the last few years in the context of SmB_6 . Indeed, according to the second sentence in the second paragraph, SmB_6 can be classified as a strongly correlated material, which under the application of external pressure exhibits a crossover from semiconducting-like to metallic-like in resistivity. This is an example of how one can control electronic topology in strongly correlated systems. The authors are undoubtedly aware of the ongoing debate on whether SmB_6 hosts topologically protected metallic surface states and that (i) there is convincing experimental evidence that this is indeed the case and (ii) numerous attempts to probe the evolution of the surface states under the influence of disorder, magnetic field and/or pressure. That is why I would like to know why the authors do not cite any papers on SmB_6 which are clearly relevant to the subject of this paper [they have done it in their PRL paper]. Consequently, I suggest that the authors either provide the relevant citations already in the introductory part of their manuscript or change the title of their manuscript which would make analogies with the physics of samarium hexaboride much less obvious.*

The Reviewer is certainly right that SmB_6 is a strongly correlated electron system. We

also confirm that we are aware of the ongoing debate about topological surface states in this material. Whereas there is good evidence for conducting surface states, the topological nature of these states remains highly controversial. It is important to note that the closing of a Kondo insulator gap by pressure, which the Reviewer mentions, is not the “topology control” we refer to here. It is a transition of one correlated phase to another (Kondo insulator to metal), which might be *accompanied* by the disappearance of signatures of the conducting surface states. By contrast, we report the magnetic-field tuned annihilation of Weyl nodes in $\text{Ce}_3\text{Bi}_4\text{Pd}_3$, *in an essentially unchanged correlated background*. In other words, the topology is controlled *per se*, *independently of underlying correlated phases*.

In fact, we had discussed this important point quite extensively throughout the manuscript (see, e.g., abstract “the pronounced topological response ... can be *genuinely* manipulated”; lines 213-217 “Importantly, this takes place *in a background of unchanged symmetry and with the Kondo effect continuing to operate*. In other words, we have realized a controlled suppression of the Kondo-driven topological semimetal. A corollary is that we *have isolated the tuning of topology from the tuning of correlation physics* as such.”; etc.).

This *genuine* control, culminating in the *annihilation of Weyl nodes*, in an otherwise *unchanged background*, is our key result, and we are not aware of any other work where this has been demonstrated.

Changes to the manuscript: Following the suggestion of the Reviewer, we have changed the title to further stress this important point. The new title reads “*Genuine control of electronic topology in a strongly correlated electron system*”. In addition, in our introduction on heavy fermion metals (lines 32-40), we have replaced the word “electron systems” by “conductors” to make it more explicit that our focus are bulk conductive, not bulk insulating states.

- 1.2 *As I am reading through the manuscript, I encounter the words like “... a plausible interpretation”, “may reflect ...” and “... likely via...”, which as a reader I interpret as if the authors are not sure what their experimental results really mean and, so after all, my impression is that author’s interpretation may well be correct, but it also may well be wrong. I am afraid this is not good enough: this is an experimental paper and the experimental data should be good enough in order to avoid being overinterpreted in one way or another.*

Generally we find a cautious language appropriate whenever new phenomena are reported. Furthermore, we would like to stress that most of the above quotations do not even indicate “hesitation in our interpretation”

- e.g., in “*A plausible interpretation*, which will receive further support from the field-dependent data presented below, is that...” it is simply explained which other experiment pins down this issue;

concern only side aspects of our work

- e.g. “The apparent fine structure ... *may reflect* various regimes of Weyl node configurations ..., which is an interesting topic for further investigations.”;

or even the work of others

- e.g. “... (we note that in a related pressure-induced transition in SmB₆ no such effects were seen^{30,31}, but this *may reflect* the need of extending those measurements ...).”

We went carefully through the entire manuscript, and found only one place where rewording seemed appropriate.

Changes to the manuscript: We replaced “likely” by “apparently” in lines 115-116.

- 1.3 *On page 2, in the third paragraph the authors claim that they discover the “topological quantum phase transition.” As we know, quantum phase transitions are usually described by the correlation functions, so what correlation function could be probed here? If the authors mean the trivial transition between the states with two different topological indices, than I suggest that they say so explicitly.*

In quantum phase transitions between phases that are distinguished by a Landau order parameter, the equilibrium correlation functions associated with the Landau order parameter would be the direct physical quantities to describe the transition, as the Reviewer stated. That, however, is not the case here. What happens instead is the annihilation of Weyl and anti-Weyl nodes (sources and sinks of Berry curvature). In that sense, it is a topological phase transition that does not involve any Landau order parameter. There could be some hidden equilibrium correlation functions that underlie such a topological quantum phase transition, but there are no obvious ones—and there may very well be none—that can be used.

The question on the topological indices does not have a clear-cut answer for strongly correlated (semi)metals, where topological indices remain to be identified. This, we thus wish to note, is beyond the scope of the present work.

Changes to the manuscript: We have revised the introduction to expand on this point, by changing “..., and the annihilation of the nodes in a topological quantum phase transition.” to “..., and the annihilation of the nodes. This transition is characterized by the suppression of singularities in the Berry curvature instead of a Landau order parameter; as such, it is a topological quantum phase transition.” (lines 44-45). We have adapted the following sentence (lines 45-47) for a smooth transition.

- 1.4 *In the same paragraph, the last sentence the authors claim that their discovery ‘opens up entirely new regime in the correlation topology interplay’. Again, in my opinion, this is a purely declarative statement. What do you mean by that?! Are you discovering here something fundamentally new and, so that, say, transitions in SmB6 induced by external pressure and/or magnetic field do not fall under this ‘regime of correlation topology interplay’? Please, be more specific and straightforward.*

This point relates back to 1.1 above. So yes, we do discover here something entirely new, viz. the *genuine* control of topological states in a strongly correlated system. This is very different from quantum phase transitions in strongly correlated electron systems between phases that are (primarily) identified via their nontopological characteristics, e.g. a Kondo insulator gap in SmB₆ that closes at a critical pressure.

Note, that we do see this kind of transition also in $\text{Ce}_3\text{Bi}_4\text{Pd}_3$, at the second critical field B_{c2} . Whereas this transition helps us to understand the overall picture, it is clearly not the key novelty of our work! We hope that this, together with our detailed reply in 1.1, clarifies this important point.

Changes to the manuscript: To improve the clarity of this important point, we have added the half sentence "..., and will facilitate a systematic search for new correlation-driven topological phases." (line 49) to the statement the Reviewer quotes.

- 1.5 *The paper severely lacks the numerical estimates so that the reader could at least get some idea in which regime the system is. For example, I think it would be helpful if the authors could explain whether one should consider a quasi-classical limit [when magnitude of magnetic field is much less than the effective Fermi energy] or a quantum one?*

The single ion Kondo temperature of $\text{Ce}_3\text{Bi}_4\text{Pd}_3$ is 13 K (≈ 1.1 meV; line 481). This can be taken as a measure of the effective Fermi temperature. The magnetic field where the Weyl nodes annihilate is about 9 T (≈ 0.5 meV; Fig. 4). Thus, we had provided all information for readers wanting to relate these two scales. However, the terms quasi-classical and quantum limit seem to allude to orbital effects. As explained in the Supplementary Information (Sect. XIII: Zeeman vs orbital effects of an applied magnetic field), in heavy fermion compounds the orbital field effect leading to quantum oscillations plays a negligible role compared to the Zeeman effect. Thus, we hope the Reviewer will agree with us that the distinction between these two limits is of minor relevance to our work.

Unfortunately, I cannot recommend this manuscript for publication in the present form.

We hope that our replies, and the associated changes to the manuscript, could clarify the Reviewer's questions and that he/she now finds it suitable for publication in Nature Communications.

Reviewer #2

In this work the authors study $\text{Ce}_3\text{Bi}_4\text{Pd}_3$ with magneto transport and torque magnetometry measurements. Their data indicate that the Weyl nodes in this system respond to magnetic fields by moving in momentum space and ultimately annihilating. This is an interesting and well written paper and certainly worthy of publication in Nature communications. However I have a few comments that I would like the authors to address before recommending publication.

We thank the Reviewer for this very positive assessment, and are glad to address his/her comments below.

- 2.1 *There is uncertainty about the actual ground state of $\text{Ce}_3\text{Bi}_4\text{Pd}_3$. For example, a recent work (arXiv:2110.08230v1) suggests it is a narrow-gap Kondo insulator. I would urge the authors to start with a short summary of what is known about this materials class, and describe the two competing scenarios. This should then be revisited in the discussion.*

In our opinion, there is no doubt about the semimetallic ground state of $\text{Ce}_3\text{Bi}_4\text{Pd}_3$ in zero magnetic field. It is unfortunate that the authors of arXiv:2110.08230v1 and Ref. 17 claim that $\text{Ce}_3\text{Bi}_4\text{Pd}_3$ is a Kondo insulator, though their Hall effect data clearly reveal a finite charge carrier concentration in the low-temperature limit, and their Hall and resistivity data in the relevant low-temperature regime are not thermally activated.

Changes to the manuscript: We have added a new section in the supplementary information (Sect. II, Semimetallic ground state of $\text{Ce}_3\text{Bi}_4\text{Pd}_3$ in zero magnetic field) that explains this fact and demonstrates it with a figure from that work (Fig. S1). We refer to this section in the main text (lines 55-56).

2.2 *The phenomenology described here is complex. A schematic of the band structure of $\text{Ce}_3\text{Bi}_4\text{Pd}_3$ at a couple of important temperatures (at zero field) and the authors conjecture for how the band structure changes with field (again at a couple of important temperatures) would be very useful.*

We are grateful to the Reviewer for this extremely helpful suggestion and expect this revision to improve the readability of the manuscript considerably. Note, however, that the temperature evolution of the correlation phenomena at play here (building up of a Kondo insulator gap, formation of Weyl nodes as full Kondo coherence sets in, etc.) is complex, and experimental signatures are washed out at elevated temperatures (due to intermixing with other excitations such as phonons). This is why we focused our detailed analyses in the manuscript on the effects of magnetic field tuning in the low-temperature limit. Thus, we think that sketching a zero-temperature correlated bandstructure is most useful.

Changes to the manuscript: We have included a cartoon that illustrates how the (zero-temperature, correlated) electronic bandstructure of $\text{Ce}_3\text{Bi}_4\text{Pd}_3$ changes under magnetic field tuning (Fig. S2) and accompanying text (Supplementary Sect. III “Cartoon of the correlated bandstructure of $\text{Ce}_3\text{Bi}_4\text{Pd}_3$ under magnetic field tuning”), and refer to it in the main text (lines 60-61 and 71).

Reviewer #3

The authors report a topological quantum phase transition of a strongly correlated Weyl-Kondo semimetal $\text{Ce}_3\text{Pd}_4\text{Bi}_3$ in strong magnetic fields. The phase transition contains two stages, from a low-field Weyl-Kondo semimetal with small hole-like fermi volume to an intermediate-field Kondo insulator with small electron-like fermi volume, and then to a high-field heavy fermion metal with large fermi volume. The transition from the Weyl-Kondo semimetal to Kondo insulator is topological because the even part of the Hall signal vanishes at 9T and the linear-in- T^3 electronic specific heat is suppressed by magnetic fields. The authors conclude this transition as field-driven, Weyl nodes annihilation. More evidences in Hall and magnetic signals support the existence of the transition from a Kondo insulator to heavy fermion metal.

This draft contains sufficient and interesting data and I would like to support it to be published in Nature communications. Yet I suggest the authors to consider a serious revise because the current draft is difficult to follow if the readers did not read through ref. 6 and

7. Below I have some suggestions.

We are grateful to the Reviewer for his/her careful assessment of our work and for seeing the novelty and interest of our results! We are glad to address his/her suggestions for making the work more accessible to other readers.

- 3.1 *Figure 1 shows some notation like ρ_{xy}^{even} . Consider the readers cannot understand it if they did not read Ref 7. I suggest the authors to add a panel of data analysis in this figure like Fig. 3A in ref. 7.*

Changes to the manuscript: We have added a new section in the supplementary information (Sect. VI, Even-in-field Hall resistivity) that explains how to extract $\rho_{xy}^{\text{even}}(B)$ from the measured data. It includes Fig. 3A of Ref. 7 as new Fig. S5, as proposed by the Reviewer. We refer to it in the main text (line 97) and in the caption of Fig. 1.

- 3.2 *I suggest the authors to rewrite the introduction part. The first paragraph in page 3 is not helpful yet there is lack of introduction of $\text{Ce}_3\text{Pd}_4\text{Bi}_3$. The authors may consider move some sentences from SI part II to the introduction part.*

Changes to the manuscript: We have inserted a new paragraph (starting on page 2, line 50) that summarizes the key Weyl-Kondo semimetal signatures of $\text{Ce}_3\text{Bi}_4\text{Pd}_3$ and refer to the SI part II (now Supplementary Sect. I). In addition, following the suggestion of Reviewer 2 (point 2.2), we have included a cartoon that illustrates how the correlated electronic bandstructure of $\text{Ce}_3\text{Bi}_4\text{Pd}_3$ changes under magnetic field tuning. We expect the two together to considerably improve the readability of the manuscript.

- 3.3 *The spontaneous Hall effect is important for supporting the existence of Weyl fermion because it directly demonstrates the Berry curvature field. Why the intermediate-field Kondo insulator with small electron pockets has no Berry curvature field?*

The intermediate state is topologically trivial because the Weyl nodes have annihilated at the (lower) critical field B_{c1} . The spontaneous Hall effect as well as its finite field continuation, the even-in-field Hall effect, are caused by the Berry curvature divergences at the Weyl nodes. Such giant (and thus experimentally readily detectable) effects cannot be caused by other “mild” Berry curvature fields.

REVIEWERS' COMMENTS

Reviewer #2 (Remarks to the Author):

The authors have addressed my comments. I have no other comments and am happy to recommend publication.

Reviewer #3 (Remarks to the Author):

The authors have replied my question and I would like to recommend the draft to be published as it.